

# Insights into the molecular mechanisms underlying the function of lysyl oxidase like 1 in cancers

Xinmeng Wang[1,2,3,*], Xiaoyi Wang[1,2,3,*], Yihan Li[4], Dan Zhao[1,2,3], Jintao He[1,2,3], Lin Wang[4], Zhengliang Li[4] and Wei Xiong[1,2,3]

[1] Department of Biochemistry and Molecular Biology, College of Basic Medical Sciences, Dali University, Dali, Yunnan, China

[2] Key Laboratory of Clinical Biochemistry Testing in Universities of Yunnan Province, College of Basic Medical Sciences, Dali University, Dali, Yunnan, China

[3] Yunnan Provincial Key Laboratory of Entomological Biopharmaceutical R&D, College of Pharmacy, Dali University, Dali, Yunnan, China

[4] Department of Radiology, The First Affiliated Hospital of Dali University, Dali University, Dali, Yunnan, China

[*] These authors contributed equally to this work.

Corresponding authors
Zhengliang Li,
lizhengliang@dali.edu.cn
Wei Xiong, xiongwei@dali.edu.cn

## ABSTRACT

Cancer is one of the primary causes of human mortality and a significant barrier to increasing human life expectancy. The effective screening, early diagnosis, and treatment of cancer have long been clinical challenges, and thus new biomarkers or molecular targets must be identified to improve the diagnosis and treatment of cancer patients. Lysyl oxidase like 1 (LOXL1), a secreted copper-dependent amine oxidase, is commonly expressed in a variety of cell types. LOXL1 can maintain the steady state of elastin, engage in extracellular matrix (ECM) remodelling. LOXL1 has diverse biological functions, and its dysregulation is the basis of many clinical diseases. The abnormal expression or activation of LOXL1 can disrupt the cellular microenvironment, contributing to the development of various diseases, such as atherosclerosis, tissue damage, fibrosis, and cancer. Recent research has revealed that LOXL1 is often overexpressed in a majority of cancers, where it plays a role in regulating tumor growth and metastasis. However, some studies have also suggested that LOXL1 may have a tumor-suppressive function. Research has indicated that the LOXL1 protein is reduced in human renal cell carcinoma (RCC) and bladder cancer (BLCA), where it acts to suppress tumor growth. Conversely, it is upregulated in human salivary adenoid cystic carcinoma (SACC), non-small cell lung cancer (NSCLC), pleural mesothelioma (PM), brain glioma, prostate cancer (PRAD), gastric cancer (GC), breast cancer (BC), thyroid carcinoma (THCA), pancreatic adenocarcinoma (PAAD), and osteosarcoma (OS). The expression of LOXL1 in colorectal cancer (CRC) remains a topic of debate, as it may either be upregulated or downregulated. These findings imply that LOXL1 may have a dual role in cancer, either inhibiting or facilitating carcinogenesis. This article provides a comprehensive review of the structure and function of LOXL1, along with its associations with cancer. Additionally, it explores the role of LOXL1 in tumor microenvironment remodeling, tumorigenesis, metastasis, and the molecular mechanisms that underpin these processes.

## INTRODUCTION

Cancer is the second leading cause of death among human diseases and the largest cause of death for people under 85 years old (*Siegel, Giaquinto & Jemal, 2024*). At present, most cancers are difficult to cure and are in great need of effective treatment methods. Over the past few centuries, many rationally designed cancer treatments have been developed. Many cancer therapies introduced long ago are still available and are still used in oncology treatment (*Sonkin, Thomas & Teicher, 2024*). From 1970 to 2023, the main treatment modalities for cancer included surgery, radiation therapy, chemotherapy, allogeneic hematopoietic stem cell transplantation, pharmacological hormone therapy, treatments targeting genes with oncogenic alterations and related signaling pathways, photodynamic therapy, antibody-drug conjugates, immune checkpoint inhibitors, bispecific T-cell engagers, oncolytic virus therapy, and chimeric antigen receptor T cell therapy. In the future, these treatment approaches will continue to be utilized (*Sonkin, Thomas & Teicher, 2024*).

The lysyl oxidase (LOX) family is a group of secreted copper-dependent amine oxidases consisting of five members, namely, LOX and lysyl oxidase like 1-4 (LOXL1-4) (*Chen et al., 2020*; *Laczko & Csiszar, 2020*) (Fig. 1A). Members of the LOX protein family share a common conserved C-terminal domain and a variable N-terminal front region. The C-terminal domain includes copper ion binding sites, lysyl tyrosyl quinone, and cytokine receptor-like domains, which are essential for their catalytic activity in cross-linking collagen and elastin (*Ye et al., 2020*). The N-terminal sequences of LOX family members are diverse, with LOX and LOXL1 containing a pro-domain. LOXL2 to LOXL4 form a subgroup of this family that participates in cell adhesion and protein-protein interactions. In the LOX protein family, the N-terminus typically starts from the first amino acid and extends approximately 20 amino acid residues, constituting a signal peptide sequence crucial for the secretion of LOX proteins. Both LOX and LOXL1 possess leader sequences, resulting in their secretion as inactive proenzymes. Zymogens directly interact with the extracellular matrix (ECM), depositing the enzyme into elastic tissues (*Rodríguez & Martínez-González, 2019*). The catalytic domains of LOX family members are highly similar, indicating that they may function in a comparable manner (*Chitty, Setargew & Cox, 2019*). LOX family proteins are involved in the cross-linking of collagen and elastin, and play a key role in the remodeling of the extracellular matrix during development, injury, fibrosis, and the progression of cancer (*Vallet & Ricard-Blum, 2019*). LOX and LOXL1 both participate in the crosslinking of collagen and elastin proteins. Abnormal expression of LOXL1 underlies many pathological processes associated with imbalanced synthesis or degradation of the ECM. In addition, the LOX family also plays an important role in cancer drug treatment. For example, research has demonstrated that intraductal xenografts show lobular carcinoma cells rely on their own extracellular matrix and LOXL1 (*Sflomos et al., 2021*). Another

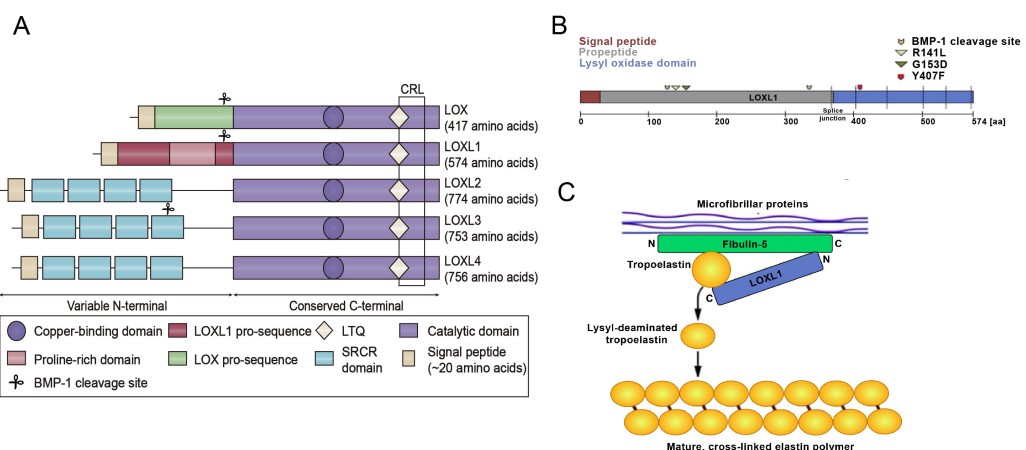

**Figure 1** **Structural homology of the family of lysyl oxidase enzymes (LOX, LOXL1-4) and the function of LOXL1.** (A) Structural homology of the family of lysyl oxidase enzymes (LOX, LOXL1-4). All LOX proteins are highly conserved in their C-terminal copper-binding catalytic domain, which is responsible for the extracellular conversion of primary amines of lysine side chains into reactive aldehydes, and are variable in their N-terminal regions, which are important for substrate binding and enzyme maturation. Both LOX and LOXL1 contain pro-sequences, which are necessary for deposition onto elastic microfibrils. These sequences are cleaved to generate catalytically active enzymes. The black square indicates the cytokine receptor-like (CRL) domain. LTQ: lysyl-tyrosyl-quinone co-factor. (B) Schematic illustration of the LOXL1 protein with the signal peptide (red), the pro-peptide (grey) and the lysyl oxidase domain (blue) indicating positions of BMP-1 cleavage sites, common variants R141L and G153D, and the rare variant Y407F. (C) Simplistic model illustrating the function of LOXL1 during elastogenesis. The N-terminal region of LOXL1 binds to the C-terminal domain of fibulin-5 to activate tropoelastin by converting it to its lysyl-deaminated form. Subsequently, covalent cross-linking of lysyl-deaminated tropoelastin occurs spontaneously to form mature, cross-linked elastin polymers.

study showed that a new ECM-targeting nanotherapy was developed by attaching an inhibitor of the ECM-related enzyme lysyl oxidase (LOX) to a lipid-based nanoparticle. When these conjugated nanoparticles were loaded with the chemotherapy drug epirubicin, they demonstrated significantly better inhibition of triple-negative breast cancer (TNBC) cell growth in both *in vitro* and *in vivo* settings. Additionally, the *in vivo* results indicated extended survival, reduced cytotoxicity, and improved biocompatibility when compared to both free epirubicin and epirubicin-loaded nanoparticles (*De Vita et al., 2021*).

Cancer research has increasingly leveraged transcriptomic data thanks to advancements in high-throughput technologies. Transcriptomics offers detailed insights into gene expression and regulation, enhancing our understanding of cancer biology (*Liu et al., 2025*). There are currently many open transcriptomic datasets available for cancer research, with The Cancer Genome Atlas (TCGA) being one of the most commonly used data mining databases. The TCGA project is an important research initiative that has provided the scientific community with a large amount of genomic and transcriptomic data. TCGA is a groundbreaking cancer genomics initiative that has molecularly analyzed more than 20,000 primary cancer samples and corresponding normal samples across 33 cancer types. This project was collaboratively initiated by the National Cancer Institute (NCI) and the National Human Genome Research Institute (NHGRI) (*Liu et al., 2025*). The

insights provided should assist the scientific community in navigating the complexities of transcriptomic data and improving the translation of research findings into clinical applications, ultimately contributing to better cancer diagnosis, treatment, and prognosis.

Given the broad range of biological functions of the LOX family, disruptions in LOXL1 expression are implicated in the development of various pathological processes associated with imbalances in ECM synthesis and degradation. This article primarily examines the structure, function, and role of LOXL1 in cancer initiation and progression. Additionally, it explores the potential applications of LOXL1 in cancer research, offering theoretical insights and references that may support clinical diagnosis and treatment strategies for cancer.

### The intended audience

This review is primarily aimed at researchers in the fields of cancer and the lysyl oxidase (LOX) family. The article focuses on examining the structure and function of LOXL1, its connection to cancer, and its involvement in tumor microenvironment remodeling, tumorigenesis, metastasis, along with the underlying molecular mechanisms.

### Survey methodology

We conducted a comprehensive literature search on PubMed and Web of Science using the keywords "LOXL1" and "Cancer". These terms were combined with Boolean operators to retrieve all types of literature related to the keywords. A total of 76 articles published between January 2011 and December 2024 were selected for this review.

## THE STRUCTURE AND FUNCTION OF THE LOXL1 PROTEIN

### Structure of the LOXL1 protein

The LOXL1 gene is situated on chromosome 15q24.1 and encodes a protein weighing 63 kDa, composed of 574 amino acids. LOXL1 expression is most prominent in the aorta, placenta, skeletal muscle, kidneys, and pancreas, suggesting its important role in preserving the structural integrity of these tissues (*Bernstein, Ritch & Wolosin, 2019*). LOXL1 is initially synthesized as a preproenzyme, which, after the removal of the N-terminal signal peptide, is secreted into the extracellular space as a proenzyme (*Laczko & Csiszar, 2020*). After the cleavage of its N-terminal signal peptide, LOXL1 is released from the cell as a proenzyme. The secreted pro-LOXL1, through its N-terminal propeptide domain, binds to the C-terminus of tropoelastin and fibulin-5, facilitating its localization to regions where elastic fiber formation takes place (*Trackman, 2018*). The function of Fibulin-5 is to bind pro-LOXL1 with tropoelastin (*Callewaert et al., 2013*). LOXL1 plays a critical role in elastogenesis, with gene mutations leading to skin laxity and age-related macular degeneration (*Greene et al., 2020*). The prepeptide region of LOXL1, abundant in lysine, has a strong affinity for tropoelastin, aiding in the correct localization and function of pro-LOXL1 through protein-protein interactions. The proenzyme is then catalytically activated by cleavage through procollagen C proteinase–bone morphogenetic protein 1 (*Li et al., 2020*). Interestingly, the predicted BMP-1 cleavage site at position 134 is located near the exfoliation syndrome (XFS)-associated coding variants at positions

141 (R141L) and 153 (G153D), suggesting that changes in LOXL1 processing may play a role in the pathogenesis of XFS (*Aung, Chan & Khor, 2018*) (Fig. 1B). Additionally, DNA methylation, long noncoding RNAs, microRNAs, and histone modifications are involved in the regulation of LOXL1 gene expression. *Zeltz et al. (2019a)* reported that LOXL1 is regulated by integrin α11. Additionally, environmental factors like hypoxia, oxidative stress, and UV radiation can influence LOXL1 expression. The interaction of these genetic and environmental factors may also impact the progression of various diseases.

## Functional significance of the LOXL1 protein

Elastic fiber formation involves tropoelastin globules being deposited onto a microfibrillar scaffold, where they accumulate and form elastic fibers through LOX and LOXL1 cross-linking (*Northington, 2011*) (Fig. 1C). Elastic fibers consist of a central core of cross-linked elastin, encircled by a network of microfibrils that contain fibrillin (*Papke & Yanagisawa, 2014*). In addition to (tropo)elastin and fibrillin-1 (FBLN-1), over 30 other ancillary proteins, including fibulins, emilins, microfibrillar-associated proteins, and LTBPs, play key roles in the assembly of elastic fibers. Specific binding interactions between fibulin-4 (FBLN-4) and LOX, as well as between fibulin-5 (FBLN-5) and LOXL1, direct both enzymes to the sites of elastic fiber assembly, regulate their oxidase activity, and aid in cross-linking. These cross-links provide elastic fibers with mechanical stability, exceptional durability, and resistance to proteolytic degradation.

The LOXL1 protein plays a key role in the maturation of elastin and the remodeling of the extracellular matrix (ECM) during tissue injury, fibrotic diseases, and the progression of malignant tumors. Its primary mechanism involves the aggregation of soluble elastin monomers into elastin polymers (*Callewaert et al., 2013*). Recent studies have shown that LOXL1 plays a role in collagen formation, assisting in maintaining the stability of the ECM (*Migulina et al., 2022*; *Trackman, Peymanfar & Roy, 2022*). In human glaucomatous trabecular meshwork cells, increased LOXL1 activity contributes to elastin matrix accumulation (*Vallabh et al., 2022*). *In vitro* cultures of human Tenon capsule fibroblasts obtained from patients with pseudoexfoliation syndrome indicated that the increased expression of elastin is related to the level of LOXL1 (*Bernstein, Ritch & Wolosin, 2018*). Additionally, an *in vitro* study of rat foetal lung fibroblasts suggested that LOXL1 combines with the elastin matrix during the process of elastin deposition. LOXL1 has also been shown to be closely linked to the function of elastic fibers in the body. Mice lacking the LOXL1 gene exhibit a range of abnormalities in elastic tissues, such as increased skin laxity, abdominal aortic aneurysms, intestinal diverticula, pulmonary emphysema, and prolapse of the pelvic cavity and genital tract (*Allen-Brady, Bortolini & Damaser, 2022*; *Jameson et al., 2020*; *Li et al., 2021*; *Meza et al., 2020*). Furthermore, recent research has shown that LOXs, including LOX and LOXL1–4, are crucial for tumour metastasis. LOXs, including LOXL1, catalyse the oxidation of lysine in collagen and preserve extracellular stiffness. In addition, LOXs, including LOXL1, contribute to tumour metastasis by generating hydrogen peroxide ($H_2O_2$), activating Src/FAK signalling and epithelial–mesenchymal transition (EMT) (*Kamiya, 2022*). In addition, *Wang et al. (2022a)* documented that the primary role of the LOX family is to remodel the tumour microenvironment (TME), and these proteins are

broadly engaged in tumour invasion and metastasis, immunomodulation, proliferation, apoptosis, and other processes. These phenomena underscore the crucial role of the LOXL1 protein in ECM homeostasis, particularly in the maturation of elastin. Other functions of the LOXL1 protein, such as the inhibition or promotion of cancer occurrence, have also been proposed. Moreover, LOXL1 can affect the tumour prognosis by affecting immune cell infiltration (*Zheng et al., 2024*). Further elucidation of the functions and mechanisms of LOXL1 could provide new insights into cancer treatment.

## Cellular signalling regulation of the LOXL1 protein

Several intracellular signaling pathways promote the secretion of LOXL1. The activity of LOXL1 mRNA is primarily induced by TGF-β1 and TGF-β2 through three distinct pathways. The TGF-β ligands bind to cell surface receptors, leading to the phosphorylation of the Smad2/Smad3 complex, which, with the help of Smad4, translocates to the nucleus (*Sethi, Wordinger & Clark, 2013*; *Sethi et al., 2011*). This complex can then bind directly to the LOXL1 promoter or interact with activator protein 1 (AP-1) to regulate transcription (*Sethi et al., 2011*). Additionally, TGF-β receptor activation can stimulate LOXL1 *via* the MAPK/JNK1/2 pathway, which phosphorylates the c-Jun component of AP-1 (*Sethi, Wordinger & Clark, 2013*; *Sethi et al., 2011*). Lastly, TGF-β may also enhance LOXL1 expression through the PI3K/Akt pathway (*Gao et al., 2018*) (Fig. 2). Furthermore, the BMP antagonist gremlin has been shown to indirectly increase LOXL1 expression by inhibiting BMP4 and BMP7, which usually suppress TGF-β2-induced LOXL1 expression (*Sethi, Wordinger & Clark, 2013*).

# THE ROLE OF LOXL1 PROTEIN IN THE OCCURRENCE AND PROGRESSION OF CANCER

Studies have shown that LOXL1 protein expression varies across different cancer tissues and cell lines. When compared to adjacent or normal tissues, LOXL1 expression is notably reduced in renal cell carcinoma (RCC) and bladder cancer (BLCA) but significantly elevated in salivary adenoid cystic carcinoma (SACC), non-small cell lung cancer (NSCLC), malignant pleural mesothelioma (MPM), brain glioma, prostate cancer (PRAD), gastric cancer (GC), breast cancer (BC), thyroid carcinoma (THCA), pancreatic adenocarcinoma (PAAD), and osteosarcoma (OS). However, the expression of LOXL1 in colorectal cancer (CRC) remains controversial, with reports suggesting both upregulation and downregulation (*Ye et al., 2020*; *Barker, Cox & Erler, 2012*) (Table 1).

The tumor-promoting mechanism of LOXL1 can be categorized into two main effects: extracellular and intracellular: (1) Extracellular effect: TGF-β secreted by non-small cell lung cancer (NSCLC) cells promotes the expression of integrin α11 (ITG α11) in cancer-associated fibroblasts (CAFs) through the Smad signaling pathway, which in turn upregulates LOXL1 expression in CAFs. This process induces ECM remodeling, fostering the proliferation and invasion of NSCLC cells. Additionally, LOXL1 secreted by tumor cells interacts with other cell types in the tumor microenvironment (TME), accelerating tumor progression. In intrahepatic cholangiocarcinoma (ICC), elevated LOXL1 expression interacts with fibulin-5 (FBLN5), an extracellular matrix glycoprotein containing the

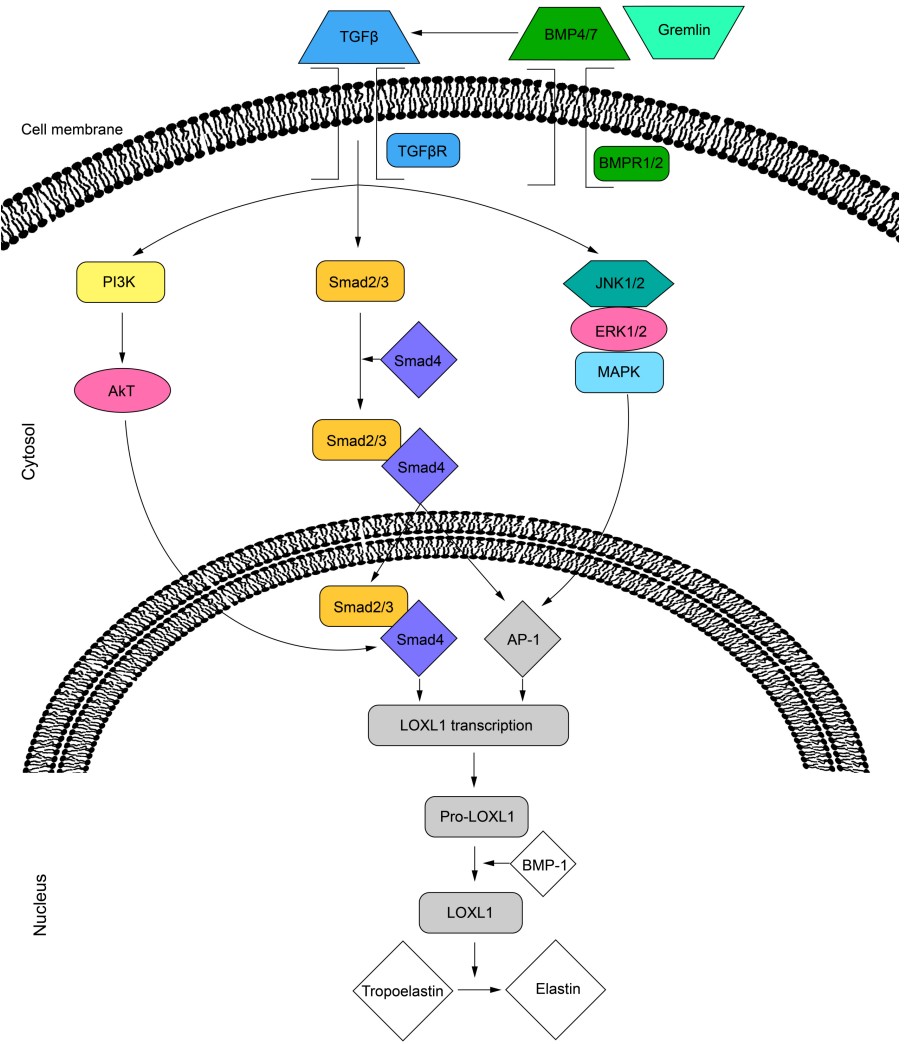

**Figure 2** **Cell signalling pathways in LOXL1 gene expression.** The proposed mechanism of TGFβ regulation of LOXL1 *via* the PI3K/AkT, Smad2/3 and MAPK/JNK1/2 pathways. TGFβ, transforming growth factor beta; TGFβR; transforming growth factor beta receptor; BMP4/7, bone morphogenic protein 4/7; BMPR1/2, bone morphogenic protein receptor 1/2; P13K, phosphoinositide 3-kinase; Smad 2, mothers against decapentaplegic homolog 2; Smad 3, mothers against decapentaplegic homolog 3; Smad 4, mothers against decapentaplegic homolog 4; JNK1/2, c-Jun N-terminal kinase 1/2; ERK1/2, extracellular signal-regulated kinase 1/2; MAPK, mitogen activated protein kinase; AP-1, activator protein 1; LOXL1, lysyl oxidase like 1; BMP-1, bone morphogenic protein 1.

Arg-Gly-Asp (RGD) motif. LOXL1, FBLN5, and ITG-αvβ3 on the surface of vascular endothelial cells (VECs) may form a complex that regulates the FAK and MAPK pathways in VECs, promoting tumor angiogenesis (*Wang et al., 2022a*) (Fig. 3A). (2) Intracellular effect: The VEGFR/Src/CCAAT enhancer binding protein α (CEBPA) axis increases LOXL1 expression in glioma cells. LOXL1 stabilizes BAG family molecular chaperone regulator 2 (BAG2) by preventing K186 ubiquitination, thereby inhibiting tumor apoptosis (*Wang et al., 2022a*) (Fig. 3B).

**Table 1 The dual role of LOXL1 in various types of human cancers.**

| Cancer type | LOXL1 expression | Function | Mechanism | Therapeutic targets | References |
|---|---|---|---|---|---|
| Renal cell carcinoma | Decreased | Tumor suppressor | The loss of LOXL1 expression is associated with tumor occurrence | N/A | *Ben-Skowronek & Kozaczuk (2015)* and *Añazco et al. (2021)* |
| Bladder cancer | Decreased | Tumor suppressor | Silenced by promoter methylation; inhibits the Ras/ERK signalling pathway; promotes the zinc finger transcription factor SNAI2, which inhibits CDH1 and initiates EMT | Ras/ERK pathway; SNAI2; CDH1 | *Li et al. (2018a)* and *Zou et al. (2024)* |
| Salivary adenoid cystic carcinoma | Increased | Oncoprotein | Hypomethylated CpG islands | N/A | *Bell et al. (2011)* |
| Non-small cell lung cancer | Increased | Oncoprotein | LOXL1 expression regulated by Integrin α11 topromote tumorigenicity | Integrin α11 | *Zeltz et al. (2019b)* and *Lee et al. (2011)* |
| Malignant pleural mesothelioma | Increased | Oncoprotein | A novel diagnostic biomarker | N/A | *Kim et al. (2020)* |
| Brain glioma | Increased | Oncoprotein | LOXL1 promotes the proliferation through the Wnt/β-catenin signalling pathway; LOXL1 exhibits antiapoptotic activity by interacting with BAG2; inhibiting the EMT pathway | Wnt/β-catenin pathway; BAG2; CDH2/VIM/ SNAI1 and CDH1 | *Davis (2016)*, *Li et al. (2018b)*, *Yu et al. (2020)*, *Fan, Li & Liu (2024)*, *Xia et al. (2022)*, *Zhong et al. (2023)*, *Zeng et al. (2023)*, *Laurentino et al. (2022)*, *Zhao et al. (2023)*, *Kumari & Kumar (2023)*, *Wang et al. (2024)*, *Yuan et al. (2024)* and *Zhang et al. (2024)* |
| Prostate cancer | Increased | Tumor suppressor or tumor promoting factor | Following *in vitro* exposure to hypoxia | β-Aminoproprionitrile reduced tumour growth but only before the cell implantation stage | *Wang et al. (2022b)* and *Nilsson et al. (2016)* |
| Gastric cancer | Increased | Oncoprotein | LOXL1 overexpression reduces CDH1 expression; increases VIM, CDH2, SNAI2, and PLS3 expression; via the WNT/beta-catenin/cyclinD1 pathway | CDH1; VIM; CDH2; SNAI2; PLS3; WNT/beta-catenin/cyclinD1 pathway | *Kasashima et al. (2018)*, *Hu et al. (2020a)*, *Wang et al. (2021)* and *Liang et al. (2024)* |
| Breast cancer | Increased | Oncoprotein | ZEB1 potentiates the $Zn^{2+}$-mediated transcription of multiple EMT-related factor LOXL1 | ZEB1 | *Ramos et al. (2022)*, *Tian et al. (2023)*, *Hirabayashi et al. (2023)* and *He et al. (2024)* |
| Thyroid carcinoma | Increased | Oncoprotein | LOXL1 is transcriptionally controlled by FOXF2 and triggers Wnt/β-catenin signalling pathway | FOXF2, Wnt/β-catenin pathway | *Fang et al. (2024)* and *Meng et al. (2022)* |
| Pancreatic adenocarcinoma | Increased | Oncoprotein | Underpin the shaping of the TME to promote cancer growth, metastasis | N/A | *Jiang et al. (2022)* |

**Table 1** (*continued*)

| Cancer type | LOXL1 expression | Function | Mechanism | Therapeutic targets | References |
|---|---|---|---|---|---|
| Osteosarcoma | Increased | Oncoprotein | One of ODRGs | N/A | *Shao et al. (2022)* |
| | Decreased | Tumor suppressor | LOXL1 activates the phosphorylation of MST1/2, inhibiting the transcription of the YAP gene | MST1/2, YAP | *Hu et al. (2020b)* |
| Colorectal cancer | Increased | Oncoprotein | High LOXL1 expression correlates with tumor differentiation status and poorer prognosis | N/A | *Li et al. (2024)* and *Sun et al. (2021)* |

**Notes.**

SNAI2, Snail family zinc finger transcription factor 2; CDH1, E-cadherin; BAG2, BAG family molecular chaperone regulator 2; CDH2, N-cadherin; VIM, vimentin; SNAI1, Snail family zinc finger transcription factor 1; PLS3, plastin 3; ZEB1, zinc finger E-box binding homeobox 1; FOXF2, Forkhead box F2; ODRGs, osteoclasts differentiation-related genes; MST1/2, mammalian sterile 20-like kinase 1/2; YAP, Yes-associated protein; N/A, not available.

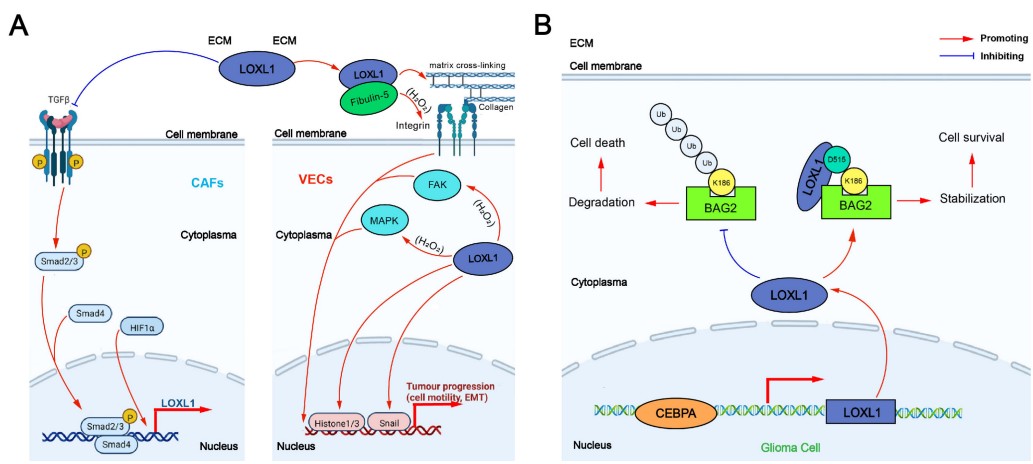

**Figure 3** **The tumor-promoting mechanism of LOXL1.** (A) Extracellular effect of LOXL1; (B) intracellular effect of LOXL1. ECM, extracellular matrix; LOXL1, lysyl oxidase-like 1; TGFβ, transforming growth factor beta; CAFs, cancer-associated fibroblasts; Smad 2/3, mothers against decapentaplegic homolog 2/3; Smad 4, mothers against decapentaplegic homolog 4; HIF1α, hypoxia inducible factor-1α; P, phosphate group; VECs, vascular endothelial cells; FAK, focal adhesion kinase; MAPK, mitogen-activated protein kinase; EMT, epithelial–mesenchymal transition; Ub, Ubiquitin; BAG2, BAG family molecular chaperone regulator 2; CEBPA, CCAAT/enhancer binding protein alpha.

The expression of LOXL1 in various human cancers was analyzed using data from The Cancer Genome Atlas (TCGA) database *via* a browser (https://portal.gdc.cancer.gov) (Fig. 4). LOXL1 transcripts are notably overexpressed in several cancers, including invasive breast carcinoma, cholangiocarcinoma, diffuse large B-cell lymphoma, esophageal carcinoma, glioblastoma multiforme, head and neck squamous cell carcinoma, kidney renal papillary cell carcinoma, lung adenocarcinoma, lung squamous cell carcinoma, ovarian serous cystadenocarcinoma, pancreatic adenocarcinoma, pheochromocytoma and paraganglioma, sarcoma, stomach adenocarcinoma, thymoma, uterine corpus endometrial carcinoma, and uterine carcinosarcoma (Fig. 4A). Furthermore, the cBioPortal database (https://www.cbioportal.org/) revealed that the LOXL1 gene frequently undergoes mutations and amplifications in tumors, with varying frequencies of gene mutations, structural

variants, amplifications, and deep deletions across different cancer types. Among these, the mutation frequency of LOXL1 is highest in uterine corpus endometrial carcinoma, while the amplification frequency is highest in uterine carcinosarcoma (Fig. 4B). Analysis of TCGA tumor samples shows that the most common point mutations in LOXL1 are D525N and R562H/C (Fig. 4C).

## Cancers exhibiting LOXL1 downregulation
### Role of LOXL1 in renal cell carcinoma

Renan cell carcinoma (RCC) is a malignant tumour originating from the epithelial cells of the renal tubules and is one of the most common tumours of the urinary system. Von Hippel−Lindau syndrome (VHL) is an autosomal dominant inherited genetic disorder in which RCC is the primary tumour type caused by VHL syndrome. Patients with VHL syndrome have an approximately 70% risk of developing RCC by the age of 60 (*Ben-Skowronek & Kozaczuk, 2015*). VHL syndrome is associated with various missense germline mutations in VHL tumour suppressor genes. The LOXL1 mRNA is downregulated in RCC cell lines with VHL gene mutations, suggesting that the loss of LOXL1 expression is associated with RCC occurrence, but the specific molecular mechanisms involved remain unclear (*Añazco et al., 2021*).

### Role of LOXL1 in bladder cancer

Promoter hypermethylation is a prevalent mechanism driving the silencing of diverse cancer genes in humans. *Li et al. (2018a)* identified 59 candidate hypermethylated genes in bladder cancer (BLCA) cells through the combination of pharmacology and microarray technology. They reported that LOXL1 is a candidate methylated gene that is frequently silenced in BLCA cell lines and that this silencing is associated primarily with promoter methylation. In primary bladder tumours, LOXL1 and LOXL4 often undergo hypermethylation, leading to a loss of expression. This study revealed the presence of somatic mutations in the LOXL4 gene in BLCA, but no mutations in the LOXL1 gene were detected. Additionally, reintroducing the LOXL1 and LOXL4 genes into human-derived BLCA cells led to a decrease in colony formation capacity. Follow-up compensation experiments indicated that elevating the expression of LOXL1 and LOXL4 genes can counteract Ras activation through the extracellular signal-regulated kinase (ERK) pathway (*Li et al., 2018a*). Therefore, this study suggested that the LOXL1 gene functions as a tumour suppressor gene by suppressing the Ras/ERK signalling pathway in BLCA. In addition, a recent study found that exposure to Benzo[a]pyrene (B[a]P)/7,8-dihydroxy-9,10-epoxybenzo[a]pyrene (BPDE) promotes the EMT process through the pivotal factor LOXL1, thereby contributing to bladder carcinogenesis (*Zou et al., 2024*). The study provides a theoretical basis for considering LOXL1 as a potential biomarker for early diagnosis and a novel target for the precise diagnosis and treatment of BLCA (*Zou et al., 2024*).

## Cancers exhibiting LOXL1 upregulation
### Role of LOXL1 in salivary adenoid cystic carcinoma (SACC)

SACC is the most prevalent malignant tumor of the salivary glands, representing roughly 22% of all malignant salivary gland neoplasms. High methylation of CpG islands on

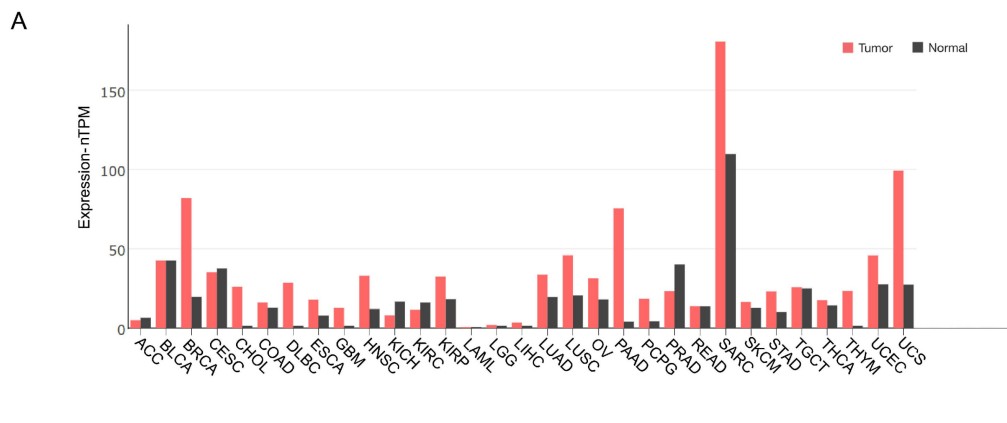

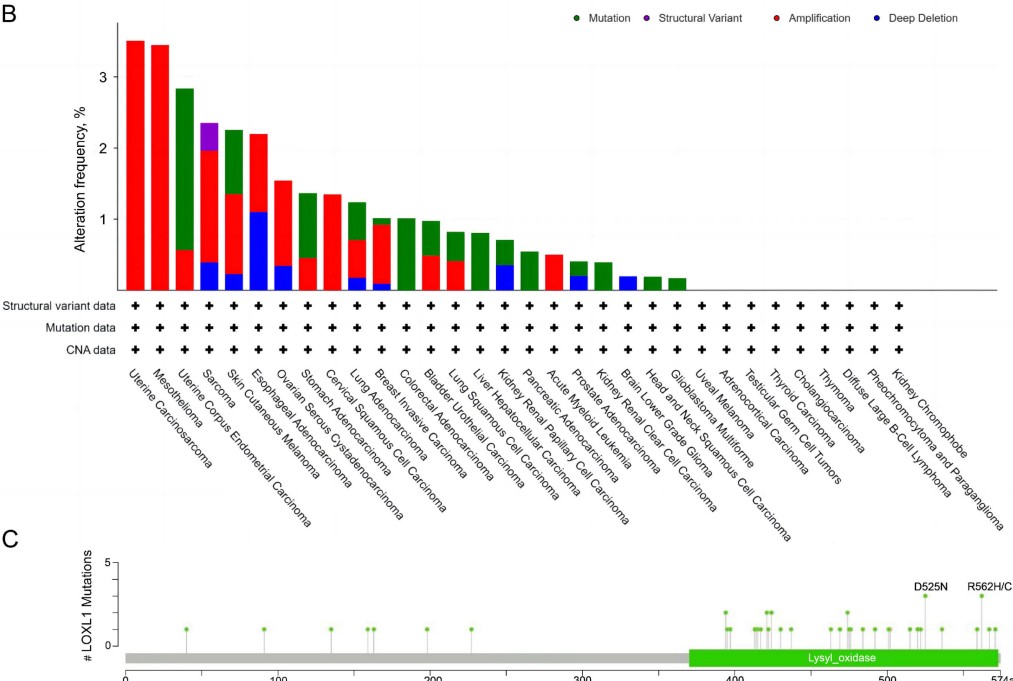

**Figure 4** **Tissue-specific expression analysis of LOXL1 and LOX1 genetic alteration in various tumor types in TCGA.** (A) Expression of LOXL4 in diverse human cancers (including tumor and normal samples). Expression levels are shown in nTPM. The $X$-axis shows normal tissue and cancer types. The $Y$-axis shows the LOXL1 mRNA expression level. (B) Differences in the frequency of mutation, amplification and deep deletion of the LOXL1 gene in human cancers. The $X$-axis shows different cancer types. The $Y$-axis shows the alteration frequency (%) of different cancer types. (C) The position of LOXL1 mutation sites are derived from cBioPortal. D525N and R562H/C are the most prevalent point mutations in LOXL1. The $X$-axis shows amino acids of LOXL1. The $Y$-axis shows the position of LOXL1 mutation sites. LOXL1, lysyl oxidase-like protein 1; nTPM, normalized transcripts per million; ACC, adrenocortical carcinoma; BLCA, bladder urothelial carcinoma; BRCA, breast invasive carcinoma; CESC, cervical squamous cell carcinoma and endocervical adenocarcinoma; CHOL, cholangio carcinoma; (continued on next page...)

**Figure 4 (…continued)**
COAD, colon adenocarcinoma; DLBC, lymphoid neoplasm diffuse large B-cell lymphoma; ESCA, esophageal carcinoma; GBM, glioblastoma multiforme; HNSC, head and neck squamous cell carcinoma; KICH, kidney chromophobe; KIRC, kidney renal clear cell carcinoma; KIRP, kidney renal papillary cell carcinoma; LAML, acute myeloid leukemia; LGG, brain lower grade glioma; LIHC, liver hepatocellular carcinoma; LUAD, lung adenocarcinoma; LUSC, lung squamous cell carcinoma; OV, ovarian serous cystadenocarcinoma; PAAD, pancreatic adenocarcinoma; PCPG, pheochromocytoma and paraganglioma; PRAD, prostate adenocarcinoma; READ, rectum adenocarcinoma; SARC, sarcoma; SKCM, skin cutaneous melanoma; STAD, stomach adenocarcinoma; TGCT, testicular germ cell tumors; THCA, thyroid carcinoma; THYM, thymoma; UCEC, uterine corpus endometrial carcinoma; UCS, uterine carcinosarcoma; CNA, copy number alterations.

gene promoters leads to transcriptional suppression. *Bell et al. (2011)* analysed abnormal DNA methylation in 16 paired normal tissues and tumour tissues and identified 32 hypermethylated and seven hypomethylated differentially methylated CpG islands in SACC tissues through a microarray analysis. CpG islands exhibit low methylation near the LOXL1 gene and four other genes. High levels of methylation were detected near 14 genes that primarily encode transcription factors and 13 genes with different functions. Additionally, compared with normal tissues, the highly methylated genes in tumour tissues are associated with development, apoptosis, and other fundamental cellular pathways, suggesting that the downregulation of these genes is related to the development and progression of SACC. Therefore, the upregulation of LOXL1 may be related to tumour development and progression. CpG methylation plays a fundamental role in a range of biological processes, including ageing, infectious diseases, and cancer. Assessing the CpG island methylation status of cancer-associated genes can offer a theoretical basis and inform the clinical diagnosis and treatment of SACC (*Bell et al., 2011*).

### Role of LOXL1 in non-small cell lung cancer

Non-small cell lung cancer (NSCLC) is a type of lung cancer that constitutes approximately 80% of all lung cancer cases. Integrin α11, a stromal collagen receptor, promotes the growth and metastasis of NSCLC and is involved in regulating collagen within the tumor's extracellular matrix (ECM). *Zeltz et al. (2019b)* found that LOXL1 expression is reduced in mice lacking integrin α11. A comprehensive analysis of patient data further explored the relationship between LOXL1 and integrin α11, revealing a correlation between their expressions. The study confirmed that the absence of integrin α11 leads to a downregulation of LOXL1 expression in stromal cells. *In vitro* experiments involving LOXL1 knockout or overexpression in CAFs demonstrated the role of LOXL1 in reshaping the collagen matrix and aligning collagen fibers, while *in vivo* studies using NSCLC xenograft models validated the tumorigenic effect of LOXL1. The results of collagen remodelling in the tumour stroma of NSCLC demonstrate that the LOXL1 protein promotes NSCLC growth and progression. Under the regulation of integrin α11 expression, stromal LOXL1 serves as a determinant of NSCLC occurrence and could be a therapeutic target for NSCLC. *Lee et al. (2011)* reported that, compared with the primary site of NSCLC, LOXL1, monocarboxylate transporter 1/2 (MCT1/2), and matrix metalloproteinase 2/9 (MMP2/9) are overexpressed at the metastatic site. Overexpression of the LOXL1 gene induces lung cancer cell metastasis

both *in vivo* and *in vitro*, while knocking down the LOXL1 gene with an siRNA can inhibit lung cancer cell metastasis and invasion. MCT1/2 reduces the pH of the acidic extracellular environment by inducing lactate accumulation and substantially increases MMP2/9 protein expression, thus participating in LOXL1-associated cancer metastasis. This study suggests that the LOXL1 protein can serve as a potential drug target for inhibiting malignant tumour metastasis.

### Role of LOXL1 in pleural mesothelioma

Pleural mesothelioma (PM) is a highly aggressive tumour that occurs in the pleural mesothelial tissue covering the lungs. The main cause of PM is exposure to asbestos in industrial environments. *Kim et al. (2020)* conducted a bioinformatics screen using a public tumour database and identified several candidate genes that could serve as biomarkers for PM. Through quantitative polymerase chain reaction, potential tumour markers in nine normal pleural tissue samples and 40 PM tissue samples were analysed. These results indicate that LOX, LOXL1, LOXL2, and zinc finger protein, FOG family member 2 (ZFPM2) are possible diagnostic markers for PM. The investigation also revealed that the LOX protein family and ZFPM2 have the same diagnostic capabilities as fibulin-3 or mesothelin (MSLN), making them more suitable potential biomarkers than cadherin 11 (CDH11), sulfatase 1 (SULF1) and thrombospondin 2 (THBS2) (*Kim et al., 2020*). Therefore, the LOX protein family and ZFPM2 can serve as innovative diagnostic markers for PM, providing more favourable references for the clinical diagnosis of PM (*Kim et al., 2020*). In recent studies using RT-qPCR, we found that the mRNA expression levels of LOXL1 and LOXL2 are significantly higher in 12 PM tissues compared to their matched normal pleural tissues. The increased expression of LOXL1 and LOXL2, along with tumor type, are independent factors associated with a poor prognosis for PM patients. Additionally, the expression of the LOXL1 gene is significantly positively correlated with the expression of the LOXL2 and LOXL4 genes.

### Role of LOXL1 in brain glioma

Brain gliomas account for around 70% of primary malignant brain tumors in adults and are recognized for their high recurrence rate and poor 5-year survival prognosis. Due to the presence of the blood–brain barrier, the range of effective clinical treatments for brain gliomas is currently limited to temozolomide (TMZ) chemotherapy and ionizing radiation (IR) (*Davis, 2016*). Glioma cells often display anti-apoptotic properties, which render existing treatment strategies ineffective. Consequently, targeting anti-apoptotic factors may offer a promising strategy to enhance the survival rate of glioma patients. Research has shown a strong connection between LOX family proteins and the initiation and progression of brain gliomas. *Li et al. (2018b)* analyzed the expression of the LOXL1 gene in 30 paired glioma and normal brain tissues and found that LOXL1 expression was significantly higher in malignant glioma samples. They further demonstrated that LOXL1 facilitates the proliferation of U87 and U251 glioma cells through the Wnt/β-catenin signaling pathway. *Yu et al. (2020)* performed an analysis using the TCGA database, which indicated that higher expression levels of LOXL1 are linked to increased malignancy and progression of brain gliomas. They examined the role of LOXL1 in promoting glioma cell survival

and inhibiting apoptosis through gain-of-function and loss-of-function experiments in both cellular and animal models (*Yu et al., 2020*). Studies have indicated that LOXL1 exhibits anti-apoptotic activity by interacting with multiple anti-apoptotic regulators, especially BAG family molecular chaperone regulator 2 (BAG2). The interaction between LOXL1-D515 and BAG2-K186 occurs *via* hydrogen bonding, and its lysyl oxidase activity competes with K186 ubiquitination, thus preventing the degradation of BAG2 protein (*Yu et al., 2020*). Moreover, research has shown that the VEGFR/Src/CEBPA signaling pathway specifically enhances LOXL1 expression. In clinical settings, patients with elevated LOXL1 levels in their blood have been observed to have higher BAG2 protein levels in glioma tissues (*Yu et al., 2020*). Overall, LOXL1 plays a crucial regulatory role in enhancing the anti-apoptotic capability of tumor cells, and targeting LOXL1 could be a viable treatment strategy for brain gliomas. Furthermore, LOXL1 in blood may serve as a biomarker for tracking glioma progression. Three independent research groups have reported that LOXL1 is overexpressed in gliomas and has significant prognostic value regarding patient outcomes, chemotherapy, and immunotherapy response (*Fan, Li & Liu, 2024*; *Xia et al., 2022*; *Zhong et al., 2023*). *Zeng et al. (2023)* developed a six-gene risk score model composed of ANG, F5, IL1A, LOXL1, LOXL2, and STEAP3, which can help predict the prognosis and offer potential treatment targets for gliomas. Additionally, *Laurentino et al. (2022)* found that the expression of the LOX family, including LOXL1, increases with the severity of astrocytoma malignancy, with the highest expression observed in glioblastoma (GBM). *Zhao et al. (2023)* further reported that neurotrophic factor-related genes (NFRGs) EN1 and LOXL1 could act as shared therapeutic targets for personalized immunotherapy in patients with Parkinson's disease (PD) and glioblastoma multiforme (GBM). Additionally, LOXL1 is associated with poor prognosis in glioblastoma multiforme patients (*Kumari & Kumar, 2023*). A recent study identified LOXL1 as a key risk gene that accelerates glioblastoma (GBM) progression, potentially through complex intercellular communication (*Wang et al., 2024*). LOXL1 expression was linked to tumor invasion and immune cell infiltration, including B cells, neutrophils, and dendritic cells. Knockdown of LOXL1 suppressed GBM cell proliferation and invasion by inhibiting the epithelial-mesenchymal transition (EMT) pathway, which involved downregulating N-cadherin (CDH2), Vimentin (VIM), and Snail family zinc finger transcription factor 2 (SNAI2), while upregulating E-cadherin (CDH1) (*Yuan et al., 2024*). Furthermore, *Zhang et al. (2024)* investigated the role of cell-in-cell (CIC)-related genes in glioblastoma (GBM) using bioinformatics and experimental approaches. Their findings identified LOXL1 as a key CIC-related gene (CICRG) with an important role in GBM.

### Role of LOXL1 in prostate cancer

Prostate cancer (PRAD) is a common malignant tumor in the male genitourinary system and ranks as the second most frequent cancer among men worldwide, representing a significant health threat (*Wang et al., 2022b*). LOX and LOXL1 are crucial enzymes involved in the deposition and maturation of the extracellular matrix (ECM). *Wang et al. (2022b)* found that LOX family proteins play a dual role in promoting the progression and metastasis of PRAD while also exhibiting tumor-suppressive effects. Their study revealed

that silencing LOXL1 leads to enhanced invasion and metastasis of prostate cancer cells, as demonstrated through genome-wide CRISPR-Cas9 screening (*Wang et al., 2022b*). Additional research showed that implanting rat prostate cancer AT-1 cells *in situ* increased the expression of LOX and LOXL1 mRNAs in both tumor and adjacent benign prostate tissues, suggesting that these enzymes may contribute to PRAD progression (*Nilsson et al., 2016*). *In vitro* experiments with AT-1 cells cultured for 24 h under normoxic and hypoxic conditions revealed significantly higher levels of LOX, LOXL1, and LOXL2 mRNAs in the hypoxic environment. This suggests that the elevation of these LOX family enzymes in both malignant and non-malignant prostate tissues may be driven by hypoxia (*Nilsson et al., 2016*). Inhibiting LOX protein expression with β-aminopropionitrile (BAPN) prior to AT-1 cell implantation suppressed tumor growth, whereas initiating BAPN treatment after tumor formation had no effect on tumor growth and might even promote it (*Nilsson et al., 2016*). Additionally, BAPN treatment did not prevent the formation of spontaneous lymph node metastases when tumor cells were injected intravenously. Collagen is a target of LOX, and a transient decrease in the collagen fibre content can be observed in tumour tissue and tumour-adjacent prostate tissue, suggesting that early BAPN therapy is more efficient at inhibiting tumour growth than later therapy. These findings indicate that the role of LOX family proteins depends on the environment, as they exhibit both tumour-suppressive and tumour-promoting characteristics in PRAD (*Nilsson et al., 2016*).

### Role of LOXL1 in gastric cancer

Gastric cancer (GC) is a malignant cancer that presents a major threat to human life, ranking as the third leading cause of cancer-related deaths worldwide. *Kasashima et al. (2018)* explored the relationship between the clinicopathological features of 597 patients with primary GC and the LOX family members (LOXL1, LOXL3, and LOXL4) using immunohistochemical techniques. Their findings indicated that the expression of LOXL1, LOXL3, and LOXL4 correlated with T-cell infiltration, lymph node metastasis, and both lymphatic and venous invasion (*Kasashima et al., 2018*). The expression of LOXL1 was also linked to the histological subtype and growth pattern of GC *in vivo*. Moreover, patients with positive LOXL1 expression had significantly poorer overall survival compared to those with negative LOXL1 expression (*Kasashima et al., 2018*). *Hu et al. (2020a)* reported that LOXL1 is highly expressed in both GC tissues and cells and is strongly associated with a poor prognosis for patients with GC. Gene set enrichment analysis (GSEA) demonstrated that LOXL1 expression was positively correlated with genes related to the epithelial-to-mesenchymal transition (EMT) (*Hu et al., 2020a*). In comparison to normal GC tissues and cell lines, LOXL1 is overexpressed in the tumor tissues and cell lines of GC patients with peritoneal dissemination. LOXL1 overexpression reduces CDH1 expression; increases VIM, CDH2, SNAI2, and plastin 3 (PLS3) expression; and promotes the migration of GC cells. The morphology of GC cells overexpressing LOXL1 changes to a spindle-shaped morphology (*Hu et al., 2020a*). Additionally, high LOXL1 mRNA expression is associated with poorly differentiated histological types and lymph node metastasis, serving as an independent adverse prognostic factor (*Hu et al., 2020a*). *Wang et al. (2021)* found that LOXL1 mRNA expression levels are significantly higher in GC. LOXL1 may play

a pivotal role in the development of GC and could serve as a biomarker for predicting tumor prognosis, as well as a potential therapeutic target. Recently, *Liang et al. (2024)* demonstrated that LOXL1 has been identified as a potential risk prognostic biomarker for gastric cancer by promoting gastric cancer proliferation *via* the WNT/beta-catenin/cyclinD1 pathway. These findings suggest that LOXL1 primarily acts as an oncogenic factor in GC.

### Role of LOXL1 in breast cancer

Breast cancer (BC) is the most common cancer in women, with approximately 2.6 million new cases diagnosed each year. *Ramos et al. (2022)* demonstrated that LOXL1 levels are significantly higher in BC tissues compared to normal tissues. Additionally, the basement membrane (BM), which plays a critical role in cancer progression and metastasis, may serve as a strong predictor of BC. The LOXL1 gene in the BM has been identified as a promising prognostic biomarker for BC, showing differential expression and a correlation with prognosis (*Tian et al., 2023*). Furthermore, *Hirabayashi et al. (2023)* reported that zinc finger E-box binding homeobox 1 (ZEB1) enhances the $Zn^{2+}$-mediated transcription of several EMT-related factors, including LOXL1 and LOXL4. The upregulation of these factors is crucial in the invasive spread of triple-negative breast cancer cells (*Hirabayashi et al., 2023*). LOXL1 holds potential not only as a prognostic biomarker but also as a target for immunotherapy in invasive breast carcinoma (BRCA) (*He et al., 2024*).

### Role of LOXL1 in thyroid carcinoma

During the last twenty years, thyroid carcinoma (THCA) has experienced a remarkable surge on a global scale, emerging as one of the most rapidly spreading malignancies in diverse countries and regions. *Fang et al. (2024)* reported that high LOXL1 expression is linked to an advanced clinical stage and shorter overall survival of THCA patients. Diminishing LOXL1 expression inhibited cell proliferation, colony formation, migration, invasion, the EMT, and angiogenesis. LOXL1 is transcriptionally controlled by Forkhead box F2 (FOXF2) and triggers Wnt/β-catenin signalling to facilitate malignant phenotypes, EMT progression, and angiogenesis in THCA cells. *Meng et al. (2022)* found that LOXL1 expression is elevated in papillary thyroid carcinoma (PTC) tissues and may be associated with overall survival (OS) and progression-free survival (PFS) in THCA patients. LOXL1 was shown to promote the proliferation and invasion of PTC cells *in vitro*. These findings suggest that LOXL1 could serve as a potential biomarker for predicting the survival of PTC patients.

### Role of LOXL1 in pancreatic adenocarcinoma

Pancreatic adenocarcinoma (PAAD) is the most aggressive of all solid cancers, characterized by late detection and limited treatment options. The development of effective therapies has been significantly challenged due to the complex heterogeneity and unique characteristics of the tumor microenvironment (TME). Lysyl oxidases (LOXs) play a crucial role in modifying the TME, promoting cancer growth and metastasis, and influencing the tumor's response to treatment. *Jiang et al. (2022)* detected increased LOXL1 mRNA and protein expression in PAAD tissue. The LOX family, including LOXL1, might hold potential

significance in PAAD oncogenesis and could serve as prognostic biomarkers, indicating a promising area for targeted therapy (*Jiang et al., 2022*).

### Role of LOXL1 in osteosarcoma

Osteosarcoma (OS) is one of the most prevalent primary malignant bone tumors. Osteoclasts have been found to play an important role in OS development. *Shao et al. (2022)* reported that LOXL1 is one of the key osteoclast differentiation-related genes (ODRGs). LOXL1 is predominantly highly expressed in metastatic tumor cells and serves as a significant predictor of patient survival (*Shao et al., 2022*).

## Cancer exhibiting upregulation or downregulation of LOXL1
### Role of LOXL1 in colorectal cancer

Colorectal cancer (CRC), which includes both colon and rectal cancer, is a common malignant tumor. *Hu et al. (2020b)* found that LOXL1 expression is reduced in CRC tissues compared to normal colorectal tissues. *In vitro*, CRC cell lines with LOXL1 gene silencing exhibited significantly increased migration, invasion, and colony formation abilities, whereas LOXL1 gene overexpression had the opposite effect. The *in vivo* results revealed that LOXL1 overexpression in CRC cells significantly inhibited tumour growth and metastatic progression. In terms of the molecular mechanism, LOXL1 activates the phosphorylation of mammalian sterile 20-like kinase 1/2 (MST1/2) by interacting with it, thereby inhibiting the transcription of the Yes-associated protein (YAP) gene (*Hu et al., 2020b*). However, *Li et al. (2024)* identified LOXL1 as a reliable biomarker for predicting prognosis and the response to immune checkpoint blockade (ICB) therapy in CRC patients. They observed that LOXL1 expression was abnormally elevated in CRC and was associated with poorer differentiation and prognosis. *Sun et al. (2021)* analyzed data from The Cancer Genome Atlas (TCGA) and conducted an online analysis using the ULCAN database. They found that the expression levels of LOXL1 and LOXL2 were significantly higher in CRC tissues compared to normal tissues. A subgroup analysis showed no significant difference in LOXL1 expression between rectal adenocarcinoma tissues and normal tissues. These studies offer different perspectives on the role of LOXL1 in CRC, leading to varying conclusions. Due to differences in emphasis and the complex nature of cancer development, which is influenced and regulated by multiple factors, understanding the impact of LOXL1 on CRC is very important for the subsequent diagnosis and treatment.

## CONCLUSIONS AND FUTURE PERSPECTIVES

With the advancements in modern medicine, cancer treatment options, including surgery, radiofrequency ablation, hepatic artery chemoembolization, systemic chemotherapy, targeted therapy, and immunotherapy, are becoming increasingly diverse. As a result, the patient prognosis has improved. However, cancer treatment remains a challenging issue in the medical field. The LOXL1 gene serves as a potential therapeutic target and plays a significant role in the occurrence and development of various types of cancer (*Schlötzer-Schrehardt & Zenkel, 2019*). This finding may be related to the origin of the tumour cells, degree of differentiation, and genetic factors. Moreover, differences in LOXL1 expression

across different cancers may also be associated with interactions between tumour stromal cells, ECM stability, and the diverse biological activities of LOXL1 itself. LOXL1 is a crucial gene that plays a diverse and significant role in maintaining cellular homeostasis. LOXL1 participates in several fundamental processes in tumor progression, especially in invasion and metastasis, while proliferation, apoptosis, chemoresistance, and other aspects also have preliminary progress. However, more studies are still required because of the tissue-specific, spatial, and temporal differences in the expression of LOXL1.

This review highlights that LOXL1 plays a key role in crosslinking collagen and elastin through oxidation, thereby preserving the rigidity and structural integrity of the ECM. The ECM is crucial in the tumor microenvironment, influencing invasiveness and metastasis. Consequently, developing drugs that target LOXL1 may offer therapeutic potential for preventing and treating tumor metastasis. In addition, considering the pivotal role of LOXL1 in various physiological processes, its systemic inhibition will inevitably cause numerous side effects. Therefore, developing novel delivery systems, such as enzyme-catalyzed responsive nanoparticles combined with specific enzyme inhibitors and standard cancer therapeutics, should be another promising direction for future exploration (*Loser et al., 2023*). New methods for cancer prevention and metastasis control are needed. Given the shared characteristics among LOX family members and the ongoing quest for a deeper understanding of their intricate structures, it is imperative to urgently devise specialized inhibitors tailored to LOXL1. The current availability of non-selective inhibitors, such as beta-aminopropionitrile (β-APN), which target the broader LOX family, underscores the pressing need for tailored solutions specific to LOXL1 (*Zhao, Liu & Luo, 2025*).

In essence, this review not only illuminates the role of LOXL1 in cancer progression but also beckons toward a future where innovative LOXL1-targeted therapeutics may reshape the landscape of cancer treatment, offering renewed hope and prospects for improved patient outcomes (*Zhao, Liu & Luo, 2025*). This review also has certain limitations. Currently, research on the specific mechanisms through which the LOX family influences tumor progression is still insufficient, so the summary of these mechanisms in this review is not comprehensive. Further research should be conducted in this area in the future.

## ACKNOWLEDGEMENTS

We greatly appreciate the hard work of our editors and reviewers. We appreciate Dr. Xing-guo Liu for checking the sentences in the article. AI tool DeepSeek was used to check the grammar of the article.

### Funding

The present study was funded by the National Natural Science Foundation of China (82160516, 32160167), Yunnan Provincial Applied Basic Research Program General Project (202201AT070004, 202301AT070023), Key Program of Yunnan Provincial Applied Basic Research Program (202001BB0500080), Yunnan Provincial Ten Thousand Talent Program

(2019), Yunnan Provincial Local University Joint Project (202101BA070001-128), Scientific Research Fund of Yunnan Provincial Department of Education (2022J0688, 2022J0716 and 2022Y808), and the Dali City Science and Technology Planning Project Support (2021KBG032), Open Project of Yunnan Provincial Key Laboratory of Entomological Biopharmaceutical R&D (AG202203, AP2022006), The Second Batch of Discipline Construction Projects of the First Affiliated Hospital of Dali University (DFYXK2023019), and the Xing-guo Liu Expert Workstation of Dali Bai Autonomous Prefecture (202402). The funders had no role in study design, data collection and analysis, decision to publish, or preparation of the manuscript.

### Grant Disclosures

The following grant information was disclosed by the authors:
The National Natural Science Foundation of China: 82160516, 32160167.
Yunnan Provincial Applied Basic Research Program General Project: 202201AT070004, 202301AT070023.
Key Program of Yunnan Provincial Applied Basic Research Program: 202001BB0500080.
Yunnan Provincial Ten Thousand Talent Program (2019).
Yunnan Provincial Local University Joint Project: 202101BA070001-128.
Scientific Research Fund of Yunnan Provincial Department of Education: 2022J0688, 2022J0716, 2022Y808.
Dali City Science and Technology Planning Project Support: 2021KBG032.
Open Project of Yunnan Provincial Key Laboratory of Entomological Biopharmaceutical R&D: AG202203, AP2022006.
The Second Batch of Discipline Construction Projects of the First Affiliated Hospital of Dali University:  DFYXK2023019.
The Xing-guo Liu Expert Workstation of Dali Bai Autonomous Prefecture: 202402.

### Competing Interests

The authors declare there are no competing interests.

### Author Contributions

- Xinmeng Wang conceived and designed the experiments, performed the experiments, prepared figures and/or tables, and approved the final draft.
- Xiaoyi Wang conceived and designed the experiments, analyzed the data, prepared figures and/or tables, authored or reviewed drafts of the article, and approved the final draft.
- Yihan Li analyzed the data, prepared figures and/or tables, authored or reviewed drafts of the article, and approved the final draft.
- Dan Zhao performed the experiments, prepared figures and/or tables, and approved the final draft.
- Jintao He performed the experiments, prepared figures and/or tables, and approved the final draft.
- Lin Wang conceived and designed the experiments, authored or reviewed drafts of the article, and approved the final draft.

- Zhengliang Li conceived and designed the experiments, authored or reviewed drafts of the article, and approved the final draft.
- Wei Xiong conceived and designed the experiments, authored or reviewed drafts of the article, and approved the final draft.

**Data Availability**

This is a literature review.

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
