# Peer review of "Insights into the molecular mechanisms underlying the function of lysyl oxidase like 1 in cancers"

_PeerJ, doi:10.7717/peerj.19628_

## Round 0.1 · original submission · Major Revisions

Your manuscript was considered interesting by the reviewers however they had a number of concerns that need to be addressed. First, they suggested that your introduction include more detail in order to more clearly explain the knowledge gap that your work closes and that specific references (which they indicated) discussing the bias of TGCA studies, the challenges associated with them, as well as review articles utilizing TCGA studies need to be included. Additionally, a section with future directions should be included. Several sections in the discussion are lacking sufficient references, including those on the role of LOXL1 in glioma, prostate cancer as well as that discussing the association between LOXL1 and prognosis in gastric cancer. Lastly, one of the reviewers suggested that you add a section on LOXL1 as a therapeutic target.

Please, submit a detailed rebuttal which shows where and how you have taken all comments and suggestions into consideration. If you do not agree with some of the reviewers’ comments or suggestions, please explain why. Your rebuttal will be critical in making a final decision on your manuscript. Please, note also that your revised version may enter a new round of review by the same or by different reviewers. Therefore, I cannot guarantee that your manuscript will eventually be accepted.

Reviewer 1 ·

Basic reporting

The manuscript from Wang and colleagues investigate the role of LOX1 biomarker in solid tumor.

Experimental design

The authors conducted a comprehensive literature search on PubMed and Web of Science using the
keywords "LOXL1" and "Cancer".

Validity of the findings

This comprehensive review provided a up to date overwiew of the involvment of LOX in tumor progression and valuable insight into its role as promising prognostic biomarker.

Additional comments

The work is interesting and has a good relevance to the field. The manuscript would benefit from the followings;

1. The authors should deep the review focusing also on the drug development for LOXL1 impairment.
2. The authors should included the following work for proper discussion:
Lysyl oxidase engineered lipid nanovesicles for the treatment of triple negative breast cancer. Sci Rep. 2021 Mar 3;11(1):5107. doi: 10.1038/s41598-021-84492-3. PMID: 33658580; PMCID: PMC7930284.
3. Study limitations should be included.

Reviewer 2 ·

Basic reporting

The manuscript presents an insightful review of the molecular mechanisms underlying the function of lysyl oxidase-like 1 (LOXL1) in cancers. The content aligns well with the journal's aims and scope, offering a comprehensive overview supported by various figures and experimental data. However, improvements are needed in terms of language clarity, organization, and citation completeness.

Experimental design

ok

Validity of the findings

ok

Additional comments

Introduction
Cite “Cancer statistics, 2024, 2024”. Then give intro in cancer therapy in general, cite NIH paper “Cancer treatments: Past, present, and future, 2024” (PMID: 38909530)for more information.
Line 23: The phrasing here is difficult to understand. Consider revising for clarity to ensure the content is accessible to an international audience.
Line 57-86: The introduction lacks sufficient detail. Expanding on the knowledge gap being addressed will provide better justification for the study. Cite previous review articles also used TCGA as support, such as “A pan-cancer-bioinformatic-based literature review of TRPM7 in cancers, 2022”and “Voltage-gated sodium channels in cancers, 2024” Discuss the bias from TCGA, refer to “Genetic expression in cancer research: Challenges and complexity, 2024” and “Technical and Biological Biases in Bulk Transcriptomic Data Mining for Cancer Research, 2025”
Structural and Functional Analysis
Line 114: Typo in "microfibrillar" (should be "microfibrillar"). Additionally, the sentence structure is complex and could be simplified for clarity.
Line 128: The phrase "assisting in maintaining the balance of the ECM" is awkward. Consider rephrasing for clarity.
Cell Signaling Pathways

Line 281-292: The discussion about diagnostic markers is informative but lacks sufficient references to recent studies. Additional citations would strengthen this section.
Line 294-315: The description of LOXL1's role in brain glioma is thorough but would benefit from more citations to recent literature for validation.
Cancer Progression and Therapeutic Targets

Line 348-364: The explanation of LOX family proteins in prostate cancer is clear, but the section lacks citations that support the claims made about hypoxia-driven enzyme expression.
Line 365-380: The role of LOXL1 in gastric cancer is well-detailed. However, the section would be more robust with additional references supporting the link between LOXL1 expression and poor prognosis.
Conclusion

Line 468-478: The conclusion effectively summarizes the findings but does not adequately address unresolved questions or future research directions. Adding these elements will strengthen the conclusion.
Figures and Tables

Figures 2 and 3: Ensure that all figures are properly labeled and that any abbreviations used are defined in the figure legends for clarity.

---

## Round 0.2 · accepted · Accept

Thank you for thoroughly addressing the reviewers' comments in your revised manuscript thus greatly improving it.

Reviewer 1 ·

Basic reporting

The work has been improved. The authors have addressed the issues raised. Now the manuscript could be considered for publication

Experimental design

The study is well designed and organized.

Validity of the findings

The findings are novel

Additional comments

The work is interesting and has a good relevance to the field. It should be considered for publication

Reviewer 2 ·

Basic reporting

ok

Experimental design

ok

Validity of the findings

ok

Additional comments

ok